# A Waterborne Gastroenteritis Outbreak Caused by a GII Norovirus in a Holiday Camp in Catalonia (Spain), 2017

**DOI:** 10.3390/v13091792

**Published:** 2021-09-08

**Authors:** Mònica Carol, Víctor Guadalupe-Fernández, Cristina Rius, Nuria Soldevila, Efrén Razquin, Susana Guix, Angela Dominguez

**Affiliations:** 1Sub-Directorate General of Surveillance and Response to Public Health Emergencies, Public Health Agency of Catalonia, Generalitat of Catalonia, 08005 Barcelona, Spain; monica.carol@gencat.cat; 2Epidemiological Service of Public Health Agency of Barcelona (ASPB), 08023 Barcelona, Spain; crius@aspb.cat; 3CIBER Epidemiologia y Salud Pública (CIBERESP), Instituto Salud Carlos III, 28029 Madrid, Spain; 4Medicine Department, University of Barcelona (UB), 08036 Barcelona, Spain; nsoldevila@ub.edu (N.S.); angela.dominguez@ub.edu (A.D.); 5Laboratory of Barcelona Public Health Agency (ASPB), 08001 Barcelona, Spain; erazquin@aspb.cat; 6Enteric Virus Laboratory, Department of Genetics, Microbiology and Statistics, Section of Microbiology, Virology and Biotechnology, Institute of Nutrition and Food Safety (INSA), School of Biology, University of Barcelona (UB), 08028 Barcelona, Spain; susanaguix@ub.edu

**Keywords:** norovirus, disease outbreaks, waterborne diseases, gastroenteritis

## Abstract

On 2 February 2017, Epidemiological Surveillance Services were notified of an outbreak of acute gastroenteritis (AGE) among schoolchildren who had taken part of a school trip from 30 January to 3 February 2017 at a holiday camp in Catalonia. A retrospective cohort study was performed to identify the causative agent, estimate the magnitude of the outbreak and identify its source, as well as to determine the route of transmission. Data collected by standardised questionnaires identified 41 episodes of AGE among 174 individuals who attended the camp. Cases had mainly symptoms of abdominal pain (73.8%), nausea (64.3%), vomiting (54.8%), diarrhoea (45.2%) and headache (42.9%). Consumption of water was associated with gastroenteritis (crude RR: 1.72, 95%CI: 1.01–2.92; adjusted RR: 1.88, 95%CI 1.03–3.56). NoV GII was detected in faeces (5 out of 13) and water samples. Additionally, faecal indicator bacteria and protozoa were detected in water samples. The outbreak showed a high attack rate and was caused by a natural water fountain not properly treated and not monitored for safety quality. There could have been a discharge of wastewater at a point close to the fountain; however, the source of contamination of the water could not be identified. Health education may be useful to eliminate risks associated with the consumption of untreated water from natural fountains.

## 1. Introduction

Noroviruses (NoVs) are a major cause of acute gastroenteritis (AGE) outbreaks worldwide. Approximately 457 outbreaks of norovirus illness, involving a total number of 1.125 related illnesses, occurred in the European Union (EU) over the last year [1]. The disease is commonly presented as mild and self-limiting, usually characterised by nausea, vomiting, abdominal cramping, diarrhoea and low-grade fever. The clinical syndrome typically lasts for up to 48 h for most people. Even though AGEs caused by NoV are generally mild and of short duration, illness may be severe, even fatal, especially among people clinically vulnerable, such as young children, the elderly and people with other medical illnesses. That is why, NoV infections are currently associated with considerable morbidity, mortality and healthcare costs [2]. After the introduction of rotavirus vaccines, NoV has become the leading cause of acute gastroenteritis in children medically attended [3,4,5].

The main transmission route for NoV is faecal–oral, from person-to-person after having direct contact with someone who is infected [6,7]. Similar to other enteric viruses, they may also be indirectly transmitted through the consumption of contaminated food and water [8,9]. The virus may survive long periods in surfaces and extreme temperatures (freezing temperatures and up to 60 °C), with low resistance to chlorine disinfection [10,11,12,13]. Due to its high stability and extreme infectivity to humans, indirect transmission from contaminated drinking water have been documented in previous studies [14,15,16,17,18]. Moreover, water and foodborne NoV outbreaks have been recently linked to climate change affecting the pattern of epidemic spread [6].

The impact of waterborne diseases and sanitation-related infections on mortality and global burden continue to be a major public health issue, especially on the poorest communities (low-income countries) and on vulnerable groups, such as the elderly and children under 5 years old [19]. In the European Union (EU), the burden of disease is often underestimated, mainly due to the reduced duration of the illness, as not all ill people seek for medical care, and errors in the diagnosis and notification of the cases [20]. Waterborne outbreaks have the potential to be rather large; however, sometimes there are difficulties to identify them by Public Health Agencies [15,21]. Despite the data gaps and limitations of the estimates, the burden of waterborne is considerable according to the Centre for Disease Prevention and Control [22].

Standards and regulations in water management, particularly in surveillance and control of water supply for human consumption, to prevent water-related infectious diseases have made enormous strides over the last century. In Spain, a Royal Decree (RD 140/2003) established the health criteria of the quality of the water for human consumption, ensuring population’s health and safety to all corners of the country. In Catalonia, a region of the northeast of Spain with the occurrence of NoV outbreaks is monitored by the Epidemiological Surveillance Network of Catalonia.

The aim of the present study is to describe the investigation of an outbreak caused by NoV among schoolchildren due to the consumption of contaminated drinking water at a winter holiday camp in 2017.

## 2. Methods

A retrospective cohort study was carried out among 174 participants of a school trip, to estimate the magnitude of the outbreak and to identify its source, as well as to study symptomatology, attack rates and relative risks. The study population consisted of individuals attending a private subsidised School in Barcelona, Spain, and who had taken part of a school trip to a holiday camp in Avià (Berguedà County, Catalonia, Spain) from 30 January to 3 February 2017. Epidemiological investigation included standardised questionnaires about sex, age, class, risk exposures and illness characteristics. Stool samples and environmental water samples were analysed for the presence of bacteria and viruses.

## 3. Case Reporting by the School Director

On 2 February 2017, the Epidemiological Surveillance and Response to Public Health Emergencies Service of the Public Health Agency of Catalonia was informed of several cases of AGE at a school trip. The school director reported that five students had become ill from 31 January to 2 February with nausea, vomiting, diarrhoea and abdominal pain during the holiday camp in Avià. A viral aetiology was suspected because incubation periods, clinical syndromes and the duration of the illness.

On 30 January 2017, schoolchildren arrived in the camp, each one ate the lunch brought with them and visited the Sanctuary of Santa Maria de Queralt, where they filled up their empty bottles with water to drink later, from a natural water fountain located near the sanctuary. The subsequent investigation associated the outbreak with the consumption of contaminated water or food during the trip to the Sanctuary of Santa Maria de Queralt. The place of Queralt is made up of the Sanctuary where the image of the Virgin of Queralt is venerated, and an annex building that once provided shelter and today is the restaurant and the station of the funicular that can be reached from the car park.

## 4. Epidemiological, Microbiological and Clinical Data

Two days after the outbreak started, a telephone survey was taken on sex, age, class, onset and nature of symptoms, time of day (morning AM or evening PM), recovery, water and food risk exposures (type of food), clinical history, doctor’s visit and hospital admission. Any individual who attended the primary exposure location during 30 January and 3 February 2017 (at-risk period of interest) and reported vomiting (≥2 episodes within 24 h) and/or diarrhoea (≥2 loose stools within 24 h) up to 48 h after leaving the primary exposure location, was defined as a primary case. Non-cases were defined as individuals who did not report any symptoms. Individuals in which symptoms of gastroenteritis were reported prior to 30 January 2017 were excluded from the study. Moreover, a confirmed case of acute gastroenteritis by NoV was defined as a patient with detection of NoV in faeces or with epidemiological relation with a microbiological confirmed case. Secondary cases, e.g., household contacts, were excluded from this study, as the source of infection was not related to the stay at the holiday camp, nor was it related to the exposure to potential risk factors associated with the trip.

Additionally, analysis was carried out to test for associations between potential risk factors (food, water consumption, the source of the water consumed, contact with previous cases, handling and hygiene practices in food handlers).

## 5. Microbiological Study

Surveyed individuals were contacted and asked to provide a stool sample. Stool samples were collected 3–5 days after the school trip was suspended and analysed for routine testing to diagnose the presence of enteropathogenic microorganisms at the Laboratory of Public Health Agency of Barcelona (Spain). Total nucleic acids were extracted from stool homogenates (10%, wt/vol) with BioMérieux NucliSens miniMAG system. Detection of NoV, genogrup I (GI) and genogroup II (GII), by real time reverse transcription-polymerase chain reaction (RT-qPCR) was performed according to ISO 15216-2:2013 [23]. Five days after the outbreak notification (7 February 2017), multiple water samples were collected from the natural fountain in the Sanctuary of Santa Maria de Queralt.

A standardised method to examine for microorganisms was performed by an accredited Public Health Laboratory of the Public Health Agency of Barcelona. Environmental samples were examined for microorganism counts at 22 °C (ISO 6222-1999), coliform bacteria (ISO 9308-2-2012), *Escherichia coli* (ISO 9308-2-2012), *Clostridium perfringens* (Env. Agency Blue Book-230 20) and *Enterococcus* (ISO 7899-2-2000) by culture methods.

NoV analysis in water samples was performed after concentrating 20-L of water by positively charged glass wool and polyethylene glycol precipitation and performing specific RTqPCR, as previously described [24], by the EntericVirus Research Group from the Faculty of Biology of the University of Barcelona.

## 6. Statistical Analysis

Descriptive statistics were calculated for the retrospective cohort study. The attack rate (AR) was calculated as the proportion of cases among the individuals exposed during the study period. Univariate analysis (cross tables, chi-squared test, linear trend analysis, relative risk) and multivariate analysis (Poisson regression) were performed. The threshold for significance was set at *p* < 0.05. Variables associated with the outcome at *p* < 0.20 in univariate analysis were introduced in the multivariate regression model. The final multivariate model was built using backward eliminations.

## 7. Results

### 7.1. Epidemiological Investigation

Of 174 individuals exposed (164 schoolchildren and 10 teachers), 64 participants returned the questionnaire (response rate 36.8%). The mean age of the participants was 13.7 years, and 29.7% were female. Two secondary cases were identified and were excluded from the analysis, as they did not meet the inclusion criteria. One case was a household contact who did not attend the camp and, the second case was a food handler, who was asymptomatic and with NoV stool sample negative, which turned out to be positive 14 days afterwards.

The onset of symptoms was between 31 January in the early morning in the first case and 4 February at noon in the last case (Figure 1). Cases had mainly symptoms of abdominal pain (73.8%), nausea (64.3%), vomiting (54.8%), diarrhoea (45.2%) and headache (42.9%). In this case, 41 individuals met the case definition; some students required health assistance, but none of them were submitted to the hospital.

The attack rate was 64.1% (41/64). The overall prevalence of water consumption was 53.1%. The multivariate analysis indicated that all subjects who drank water from the natural fountain had a significant increased risk of developing gastroenteritis compared with those who did not (adjusted relative risk 1.88, 95% CI: 1.03–3.56) (Table 1).

### 7.2. Environmental and Microbiological Investigations

NoV GII was detected in five of 13 faecal samples by RT-qPCR. Within laboratory examination, GII.P16_GII.2 genotype was identified in the sequence analysis.

In addition, water samples tested positive for faecal indicator bacteria, coliforms and *Escherichia coli*, negative for *Clostridium perfringens* and enterococci (Table 2). RT-qPCR tests for NoV GII were also positive in the water. However, no sequence analysis was performed in the water samples, due to their low viral loads.

## 8. Discussion

This study describes a waterborne outbreak caused by the ingestion of untreated water from a natural source of the Sanctuary of Santa Maria de Queralt, Catalonia, Spain.

Human NoV causes waterborne outbreaks worldwide suggesting their ability to persist and survive for extended periods in the environment [14,15,16,17,18,25,26,27,28,29,30,31,32]. The natural area surroundings included the Sanctuary and a restaurant that could be the source of contamination due to the wastewater discharges. A week after, there was very little rain which may have caused a decreased in ground-water level and thereby a high risk of contamination from the surface. As experts have previously reported, a false sense of security among people regarding mountain water consumption is highlighted [33].

In waterborne outbreaks, the strength of evidence involving water, as the cause of the outbreak, is determined based on the findings from epidemiological and microbiological investigations. There are difficulties in isolating NoV in water, so Tillett et al. developed a system of levels of evidence to link an outbreak to water [34]. Nevertheless, in the outbreak we are describing, NoV was isolated from the source of exposure, and it was therefore not necessary to carry out further analyses.

Both, the microbiological examination of water and stool samples, with subsequent NoV identification, and the epidemiological investigation of the outbreak strongly indicated that contamination of the water by faecal indicator bacteria was the original source of exposure [24,25,26,35]. Rapid clinical diagnosis and early control measures implemented by Epidemiological Surveillance Services prevented further transmission by person-to-person [36].

NoV GII was isolated in water as well as in stool samples, and GII.P16_GII.2 genotype was identified as the cause of AGE. Even though most waterborne NoV AGE outbreaks have been attributed to genotype GI, genotype GII have been implicated in several waterborne outbreaks, particularly GII.3, GII.6 and GII.4 [35]. Currently, genotype GII.4 is the most common cause of NoV infection worldwide [35,37,38].

Water sampling for NoV analysis was carried out five days after the water exposure, which led to the positive results. Visitors who had drunk fountain water had a significantly increased risk of AGE. Even though some authors have reported dose-response relationship in previous food and waterborne outbreaks caused by human NoV [39,40,41], the qualitative approach of this research did not allow estimating dose-response relationships between the severity and duration of the gastrointestinal disease and the consumption water [39].

Our study estimated an attack rate of acute gastroenteritis illness of 64%. The possibility of identifying the presence of the virus in the water would be related to a high concentration of the virus. The epidemic curve showed the rapid emergence of cases. According to the scientific literature, waterborne outbreaks of NoV have been often associated with high attack rates [15,42,43]. Additionally, coinfections of NoV have been reported with other pathogens and they have been associated with an increased severity of gastrointestinal disease [44,45]. In this outbreak, no other pathogen was analysed in stool samples although faecal indicator bacteria were found in water samples. Low severity symptoms suggest that coinfection may have not occurred.

The study of this outbreak highlights the need to rapidly connect an outbreak to its cause to reduce attack rates by implementing the correct measurements. No new primary cases from other groups were observed without further exacerbated by person-to-person transmission. The Epidemiological Surveillance Network and the effective collaboration between physicians, public health services, microbiologists and public water sources played a key role for the quick detection of the origin of the outbreak.

The limitations of this study are as follows. First, the qualitative and retrospective approach may involve recall bias and uncertainty in the associations’ assumptions; however, epidemiological surveys are a commonly used tool which have been widely used in the investigation of NoV outbreaks [40]. Second, the presence of faecal indicator bacteria did not exclude the possible involvement of other etiologic agents; nevertheless, given the suspicion of viral AGE, bacterial tests were not carried out. Third, some wastewater discharges as contamination source were hypothesised but not confirmed. Fourth, although no sequence analysis was performed in the water samples, epidemiologic, laboratory and environmental analysis suggested that the water from the natural fountain was the most likely vehicle for transmission. However, public health actions prevented new visitors from becoming ill and no additional outbreaks related to the natural fountain have been notified to Public Health Authorities since 2017”.

In conclusion, our results support the likelihood of waterborne transmission by a contaminated natural water fountain, identifying NoV GII.P16_GII.2 genotype as the causative agent of the outbreak. It is important to note that despite tested water was positive for GII NoV by RTqPCR, sequence-based identification could not be performed in the water samples, and nucleotide sequences were only obtained for the stool samples of those involved in the outbreak. Even though warning signs for non-potable/unfiltered/untreated water may be regularly identified in ponds, streams and natural water fountains that are often visited by the general public, the increasing frequency of waterborne NoV outbreaks call for the introduction of other approaches to prevent further outbreaks, such as periodic inspections, testing and dis-infection of the water of risk areas. Waterborne NoV outbreaks should be suspected by the high attack rates, especially when the main symptoms are vomiting or diarrhoea, and when the duration of the disease is short. Health education may be particularly useful to avoid risks associated with the consumption of untreated water from natural ponds, streams and fountains.

## Figures and Tables

**Figure 1 viruses-13-01792-f001:**
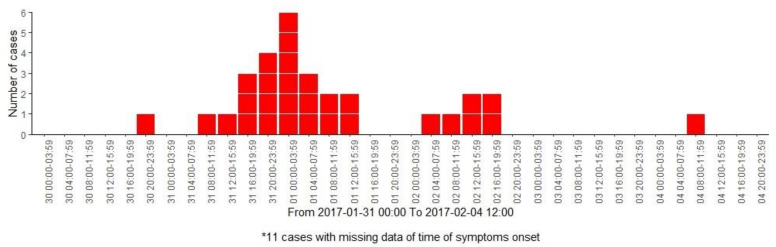
Epidemic curve of primary affected individuals by time of symptoms onset.

**Table 1 viruses-13-01792-t001:** Univariate and multivariate analyses of risk factors (food and beverages) for being a case.

Characteristic	Case	No Case	Relative Risk (Unadjusted)	*p* Value	Relative Risk (Adjusted)	*p* Value
Monday 30 January 2017						
Dry-cured ham with bread	25 (75.8%)	17 (81.0%)	0.89 (0.56–1.43)	0.65		
Bread with sausage	28 (84.8%)	16 (76.2%)	1.27 (0.66–2.46)	0.42		
Bread with chocolate	29 (87.9%)	15 (71.4%)	1.65 (0.75–3.62)	0.16		
Spaghetti	31 (91.2%)	21 (100%)	0.60 (0.18–1.95)	0.28		
Water from natural fountain near the Sanctuary	25 (73.5%)	9 (42.9%)	1.72 (1.01–2.92)	0.02	1.88 (1.03–3.56)	0.03
Cod Nuggets	28 (82.4%)	19 (90.5%)	0.79 (0.50–1.26)	0.70		
Potatoes with mushrooms	32 (94.1%)	18 (85.7%)	1.60 (0.54–4.78)	0.36		
Fruit	24 (70.6%)	16 (76.2%)	0.90 (0.58–1.39)	0.65		
**Tuesday 31 January 2017**						
Chocolate milk	25 (73.5%)	16 (76.2%)	0.95 (0.60–1.50)	0.83		
Biscuits	24 (72.7%)	13 (61.9%)	1.23 (0.74–2.03)	0.40		
Bread, butter and jam	23 (69.7%)	11 (52.4%)	1.35 (0.82–2.22)	0.20		
Lentils	32 (91.4%)	21 (100%)	0.60 (0.18–1.97)	0.28		
Pork tenderloin	33 (94.3%)	21 (100%)	0.61 (0.15–2.55)	0.52		
Yogurt	29 (85.3%)	19 (90.5%)	0.85 (0.50–1.42)	0.70		
Bread with cheese	25 (71.4%)	10 (47.6%)	1.50 (0.91–2.46)	0.08	1.54 (0.82–2.92)	0.18
Beans	27 (79.4%)	18 (85.7%)	0.86 (0.54–1.37)	0.72		
Battered fish	26 (76.5%)	15 (71.4%)	1.11 (0.67–1.85)	0.68		
Aubergine	21 (61.8%)	18 (85.7%)	0.66 (0.46–0.96)	0.05	0.38 (0.14–1.05)	0.06
Fruit	24 (72.7%)	14 (66.7%)	1.12 (0.68–1.84)	0.63		

**Table 2 viruses-13-01792-t002:** Microbial and chemical results from the natural water font. Samples were collected during the environmental investigation (February) in response to the NoV outbreak.

Parameter/Analysis	Results	Method	Limit Parametric Value
Turbidity	<0.20 FNU ^a^	MA/2/30504	1 FNU
Nitrates	<0.100 mg/L	MA/2/02002	50 mg/L
Ammonium	<0.100 mg/L	MA/2/02006	0.50 mg/L
Permanganate Oxidability	0.6 ± 0.1 mg O_2_/L	MA/2/30400 mi	5 mg O_2_/L
Microorganisms at 22 °C (total count)	5.3 × 10^2^ NMP/100 mL ^b^	ISO 6222:1999	-
Coliforms (total count)	4.1 × 10^1^ NMP/100 mL	ISO 9308-2:2012	0 CFU
*Escherichia coli* (total count)	2.0 NMP/100 mL	ISO 9308-2:2012	0 CFU
*Clostridium perfringens*	<1 CFU/100 mL ^c^	Environmental Agency Blue Book 230 (2010)	0 CFU
Enterococci	<1 CFU/100 mL	ISO 9308-2:2012	0 CFU
NoV GII	<100 copies/L	ISO/TS 15216-1:2013	0 copies/L

^a^ FNU = Formazin Nephelometric Units. ^b^ NMP = Most Probable Number. ^c^ CFU = Colony Forming Units.

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
