# Peer review of "A Waterborne Gastroenteritis Outbreak Caused by a GII Norovirus in a Holiday Camp in Catalonia (Spain), 2017"

_viruses, 2021, doi:10.3390/v13091792_

Round 1

Reviewer 1 Report

In this manuscript, Carol and colleagues investigated a norovirus outbreak among children in a holiday camp. The authors concluded that the outbreak resulted from consuming contaminated water at the camp. This a short case report that was written as a full research article. It is highly descriptive. Therefore, this reviewer does not think that the manuscript has enough justification to be published in this journal. 

Author Response

Dear Reviewer 1,

Firstly, we would like to thanks the reviewer the time dedicated to revise the manuscript.

Hereafter, we examined and commented every suggestion by the Reviewer.

Reviewer 2 Report

Manuscript Number: viruses_1308522

The manuscript by Mònica Carol et al., describes the “A waterborne gastroenteritis outbreak in a holiday camp caused by norovirus in Catalonia (Spain)”. The authors describe a retrospective cohort study and investigation of an AGE outbreak caused by norovirus water contamination in a camp in Catalonia.

Comments:

Norovirus is a leading cause of acute diarrhea disease of all population in the world. Several factors are currently increasing the challenge posed by norovirus infections to global healthy, one of them is high contagious by low dose and easily transmission through food, environmental contamination or person-to-person close contact, which complicates the healthy policy control.

This retrospective cohort study, started telephone survey and microbiology identification of fecal specimens (after 3-5 days of school trip) and water sample (from a natural fountain 5 days after outbreak notification) collected after case reporting of a school trip.

Questions:

1. In the Result:

Suggest the authors can have an epi-curve of this outbreak and indicate the onset date and identity of cases.

2. In the Method: “….each one ate the lunch brought with them and visited the Sanctuary of Santa Maria de Queralt, where they drank unsafe water.”

In the Table 1: “water from Sanctuary de Queralt”

What’s the definition of “water from Sanctuary de Queralt” in questionnaire and Table 1?  
Does it mean cases drank water or bottle water or contact the natural fountain or…? Please make a definition “water from Sanctuary de Queralt” to clarify the risk.

3. In the Method: ” The school director reported that five students had become ill from January 31th to February 2nd with…”

In the Result: NV GII was detected in 5 feces of 13 fecal sample by RT-qPCR.

Do the faces detected NV GII positive same as the illness reported? Do the positive cases drank water from Sanctuary de Queralt?  

4. In the Result: NV GII was detected in 5 feces of 13 fecal sample and water sample by RT-qPCR.

This study had limitation to illustrate the risk of “water from Sanctuary de Queralt” in lacking the connection or transmission route from water to cases, such as detected same viral strains in the same cluster in faces and water sample by sequences phylogenetic analysis. The authors can “BMC Public Health (2017) 17:870”.

Author Response

Dear Reviewer 2,

Firstly, we would like to thanks the reviewer the time dedicated to revise the manuscript.

Hereafter, we examined and commented every suggestion by the Reviewer.

Reviewer 3 Report

The manuscript by Carol et al., describe a waterborne gastroenteritis outbreak in a holiday camp and indicates as a causative agent norovirus GII; however, since no genomic sequencing was performed the authors were unable to provide evidence to establish the epidemiological link between the contaminated water with the cases of acute gastroenteritis. Therefore, I recommend that genomic sequencing be performed in the least a few water and fecal samples from the individuals from the outbreak to establish this link.

Author Response

Dear Reviewer 3,

Firstly, we would like to thanks the reviewer the time dedicated to revise the manuscript.

Hereafter, we examined and commented every suggestion by the Reviewer.

Reviewer 4 Report

A waterborne gastroenteritis outbreak in a holiday camp caused by norovirus in Catalonia (Spain)

By Monica Carol* et al (*Corresponding author)

Submitted to Viruses (Editorial No. viruses_1308522)

General Comments

This is a short retrospective report on an outbreak of acute gastroenteritis (AGE) among school children and their teachers visiting a holiday camp in Catalonia/Spain in 2017. The outbreak has been characterized by clinical symptoms, epidemiological investigations, attack rate, microbiological investigations and public health measures undertaken to limit the outbreak. Norovirus (NoV) of group II was identified as causative agent in human faecal specimens and in samples of a natural water fountain inside the camp. A significant correlation was found between the degree of drinking fountain water and the likelihood of falling ill with AGE.

Water-borne outbreaks of NoV AGE have been described before. The particular value of this report is considered to be in the joint description of rapid diagnostic measures, epidemiological investigations and preventive public health measures undertaken to limit the outbreak. Since NoV classification has been taken further [Kroneman A, et al. Proposal for a unified norovirus nomenclature and genotyping. Arch Virol. 2013 Oct;158(10):2059-68] than utilized here, it is suggested to carry out additional testing to determine the genotypes of the RNA-dependent RNA polymerase and the capsid protein of the causative agent [see also ref 22].  This would strengthen the evidence of the fountain water being the source of the outbreak; it would also help identifying whether the NoV GII is a recombinant [with recombination being a frequent event in NoV evolution]. The description of search for environmental contamination of the water fountain could be expanded.

Specific Comments

Page

1          Title, consider slight rephrasing, e.g.: ‘A waterborne gastroenteritis outbreak caused by a GII norovirus in a holiday camp in Catalonia (Spain), 2017’, or similar.                            The corresponding author should be identified by *, and the number 7 become a superscript.

            Abstract. Line 4 from bottom. Consider reading: … There could have been a discharge…

            Line 4 from bottom. … especially among people clinically vulnerable… Please clarify.

            Last line. Consider citation of:

Hemming M, et al. Major reduction of rotavirus, but not norovirus, gastroenteritis in children seen in hospital after the introduction of RotaTeq vaccine into the National Immunization Programme in Finland. Eur J Pediatr. 2013 Jun;172(6):739-46.                         

Bucardo F, et al. Predominance of norovirus and sapovirus in Nicaragua after implementation of universal rotavirus vaccination. PLoS One. 2014 May 21;9(5):e98201

2          Paragraph 1, end. … affecting the pattern of epidemic spread.

            Paragraph 2. … however, sometimes there are difficulties… Centers for Disease Control and Prevention…

            Paragraph 3. … established… northeast of Spain, the occurrence of NoV outbreaks…

            Consider omitting the last sentence.

            Line 5 from bottom. … A viral etiology was suspected because of…

            Last line.  … The subsequent investigation…

3          Microbiology study, paragraph 2. … collected from the natural fountain…

4          Paragraph 1. The epidemic curve (referred to in Discussion) should be shown.

Paragraph 2, line 1. … of symptoms was between … and …

Last paragraph. NoV GII. It is suggested to characterize the causative NoV further (see General Comments).

5          Table 2. Legend NPM or NMP? Please clarify.

            Discussion, paragraph 3, line 3. … Tillett et al  [27]. The first author is not Tillett, but Zhou X. Tillett is not a co-author.  Please clarify.

            Paragraph 5. … GEA… Spell out at first mentioning.

            Penultimate paragraph.  … the qualitative and retrospective approach…

            Last line. … The epidemic curve showed… See comment p. 4.      

6          Line 4. Ref [42] is not in the ref. list.

            Paragraph 3. … The limitations of this study are as follows.

Author Response

Dear Reviewer 4,

Firstly, we would like to thanks the reviewer the time dedicated to revise the manuscript.

Hereafter, we examined and commented every suggestion by the Reviewer.

Round 2

Reviewer 1 Report

The authors have addressed my concerns.

Author Response

Dear Reviewer 1,

We thank you again for the time dedicated reviewing this manuscript.

Conclusions were modified with the final changes.

“In conclusion, our results support the likelihood of waterborne transmission by a con-taminated natural water fountain, identifying NoV GII.P16_GII.2 genotype as the causative agent of the outbreak. It is important to note that despite tested water was positive for GII NoV by RTqPCR, sequence-based identification could not be performed in the water samples, and nucleotide sequences were only obtained for the stool sam-ples of those involved in the outbreak. Although warning signs for non-potable/unfiltered/untreated water may be regularly identified in ponds, streams and natural water fountains that are often visited by the general public, the increasing frequency of waterborne NoV outbreaks call for the introduction of other approaches to prevent further outbreaks, such as periodic inspections, testing and dis-infection of the water of risk areas. Waterborne NoV outbreaks should be suspected by the high attack rates, especially when the main symptoms are vomiting or diarrhoea, and when the duration of the disease is short. Health education may be particularly useful to avoid risks associated with the consumption of untreated water from natural ponds, streams and fountains”.

Best regards

Reviewer 3 Report

The authors have included in the manuscript the main limitation of the study which is the non-sequencing of positive water and stool samples, therefore the conclusions must be done accordingly. 

Author Response

Dear Reviewer 3,

Firstly, we would like to thanks the reviewer the time dedicated reviewing the manuscript.

Hereafter, we corrected the suggestions by the Reviewer.

Comments and Suggestions for Authors

Manuscript Number: viruses_1308522

The authors have included in the manuscript the main limitation of the study which is the non-sequencing of positive water and stool samples, therefore the conclusions must be done accordingly

Conclusions were modified with the final changes.

“In conclusion, our results support the likelihood of waterborne transmission by a con-taminated natural water fountain, identifying NoV GII.P16_GII.2 genotype as the causative agent of the outbreak. It is important to note that despite tested water was positive for GII NoV by RTqPCR, sequence-based identification could not be performed in the water samples, and nucleotide sequences were only obtained for the stool sam-ples of those involved in the outbreak. Although warning signs for non-potable/unfiltered/untreated water may be regularly identified in ponds, streams and natural water fountains that are often visited by the general public, the increasing frequency of waterborne NoV outbreaks call for the introduction of other approaches to prevent further outbreaks, such as periodic inspections, testing and dis-infection of the water of risk areas. Waterborne NoV outbreaks should be suspected by the high attack rates, especially when the main symptoms are vomiting or diarrhoea, and when the duration of the disease is short. Health education may be particularly useful to avoid risks associated with the consumption of untreated water from natural ponds, streams and fountains”.